# Safety and Tolerability of Six Months of Isoniazid Plus Pyridoxine or Three Months of Rifampicin for Tuberculosis among Subjects with Diabetes Mellitus: A Randomized Trial

**DOI:** 10.3390/microorganisms11081917

**Published:** 2023-07-28

**Authors:** Karla M. Tamez-Torres, Norma Mongua-Rodríguez, Leticia Ferreyra-Reyes, Pedro Torres-Gonzalez, Guadalupe Delgado-Sánchez, Maribel Martínez-Hernández, Miriam Bobadilla-del-Valle, Velma Y. Jasso-Sosa, Priscila del S. López-Castillo, Elizabeth Ferreira-Guerrero, Luis Pablo Cruz-Hervert, Jose Sifuentes-Osornio, Carlos A. Aguilar-Salinas, Lourdes García-García, Alfredo Ponce-de-Leon

**Affiliations:** 1Departamento de Infectología, Instituto Nacional de Ciencias Médicas y Nutrición Salvador Zubirán, Mexico City 14080, Mexico; karla.tamez@gmail.com (K.M.T.-T.); pedrotorresgonzalez@gmail.com (P.T.-G.); velmajasso@gmail.com (V.Y.J.-S.); priscila.castillochi@gmail.com (P.d.S.L.-C.); 2Instituto Nacional de Salud Pública, Cuernavaca 62100, Mexico; monguanorma@gmail.com (N.M.-R.); freyes.ld@gmail.com (L.F.-R.); gpe.delgado.s@gmail.com (G.D.-S.); lebiram1287@yahoo.com.mx (M.M.-H.); cielo2219@gmail.com (E.F.-G.); aeoorto@gmail.com (L.P.C.-H.); 3Laboratorio de Microbiología, Instituto Nacional de Ciencias Médicas y Nutrición Salvador Zubirán, Mexico City 14080, Mexico; mbv99@hotmail.com; 4División de Estudios de Posgrado e Investigación, Facultad de Odontología, Universidad Nacional Autónoma de México, Mexico City 04510, Mexico; 5Dirección de Medicina, Instituto Nacional de Ciencias Médicas y Nutrición Salvador Zubirán, Mexico City 14080, Mexico; sifuentesosornio@gmail.com; 6Unidad de Investigación de Enfermedades Metabólicas, Instituto Nacional de Ciencias Médicas y Nutrición Salvador Zubirán, Mexico City 14080, Mexico; caguilarsalinas@yahoo.com; 7Departamento de Endocrinología y Metabolismo, Instituto Nacional de Ciencias Médicas y Nutrición Salvador Zubirán, Mexico City 14080, Mexico

**Keywords:** tuberculosis infection, tuberculosis disease, tuberculosis preventive treatment, latent tuberculosis infection, diabetes mellitus, adverse events, isoniazid, rifampicin, randomized controlled trial, Mexico

## Abstract

Tuberculosis (TB) associated with diabetes mellitus (DM) is a growing problem, particularly in low- and medium-resource countries. We conducted an open-label, parallel-group, randomized, and controlled trial in a tertiary care center in Mexico City to assess TB preventive treatment (TPT) with isoniazid (INH) or rifampicin (RIF) in people with type 2 DM. Participants were assigned six months of INH 300 mg/day plus pyridoxine 75 mg or three months of RIF 600 mg/day. The primary outcomes were adverse events resulting in permanent treatment cessation and considered possibly or probably related to study drugs. We included 130 subjects, 68 randomized to INH and 62 to RIF. We prematurely halted the study based on recommendations of the Adverse Event Safety Panel. There was no difference between arms in the overall frequency of adverse events. However, the INH group had significantly more permanent treatment interruptions due to grade 2 recurrent or grade 3 or 4 hepatoxicity. In comparison, the RIF arm had more treatment interruptions due to grade 3 or 4 gastrointestinal intolerance. TPT using INH or RIF is not safe enough to be considered a universal indication to patients with type 2 DM and TB infection. These results underline the need to search for alternative TB preventions with better safety profiles for type 2 DM patients.

## 1. Introduction

The World Health Organization (WHO) estimates that one-fourth of the world population is infected with Mycobacterium tuberculosis, the microorganism that causes tuberculosis (TB). Estimates indicate that in 2021, 10.6 million people and 1.6 million fell ill and died with TB, respectively [1]. Simultaneously, the global burden of diabetes mellitus (DM) is rising. In 2017, there were an estimated 425 million people globally with type 2 DM, with numbers predicted to increase to 629 million by 2045 [2]. Many studies have investigated the interaction between DM and TB. A recent systematic review demonstrated that the risk of TB among people with DM triples that of people without DM, regardless of study design and population [3]. Patients with type 2 DM and TB have more severe clinical manifestations, a longer time to smear conversion, and a higher probability of treatment failure, mortality, and recurrence or reinfection [4,5]. Thus, the increasing burden of type 2 DM worldwide may offset the global decrease in TB incidence. Although much of the physiopathology of the association of these two diseases has yet to be understood, changes to the immune system of patients with active TB and DM have been described [6], including reductions in the activation of alveolar macrophages and in the capacity to produce IL-10 [7,8], decreases in Th1 cytokines [9,10], and abnormalities in the innate response [11]. Most studies indicate failing innate immunity but amplified adaptive immunity to Mycobacterium tuberculosis involving excess advanced glycation end products and their receptor, higher levels of reactive oxidative species and oxidative stress, epigenetic modifications due to chronic hyperglycemia, and altered nuclear receptors and or differences in cell metabolism [12].

In Mexico, TB continues to represent a public health problem aggravated by the emergence of type 2 DM and, recently, by the COVID-19 pandemic. In contrast to what is happening on a global scale, where a decrease in incidence has been observed, Mexico has experienced a 13% increase since 2015; the WHO estimates that during 2021, the TB rate was 25 (19–32) cases per 100,000 inhabitants [13]. Type 2 DM is a problem that has increased in recent decades. Adult-diagnosed DM prevalence in Mexico increased from 7% to 8.9% to 10.3% from 2006 to 2012 to 2018 [14,15]. One-fifth of patients diagnosed with TB also suffer from type 2 DM. It is likely that the increasing type 2 DM epidemic has had an impact on the rates of pulmonary TB [16].

Most *M. tuberculosis* infections can be avoided to progress to disease with tuberculosis preventive treatment (TPT) [17]. WHO’s guidelines recommend treating TB infection with 6 or 9 months of daily isoniazid (INH), a 3-month regimen of weekly rifapentine plus INH, or a 3-month regimen of daily isoniazid plus rifampicin (RIF). Alternatively, a 1-month regimen of daily rifapentine plus INH or four months of daily RIF alone may also be offered. These schedules update previous guidelines, which recommended 3 or 4 months of RIF [18]. Recommendations for TB infection among patients with DM have not been determined since there is limited information regarding efficacy and safety [2,19]. TPT is one of the critical interventions recommended by WHO to achieve the End TB Strategy targets, as upheld by the United Nations High-Level Meeting on TB in September 2018. Experts have increasingly identified the need to treat TB infection to reach the End TB Strategy targets of reducing deaths by 95% and cases by 90% by 2035 [20]. Modeling studies have concluded that treating people with TB infection is the most effective way of reducing TB incidence [21].

Given the magnitude and relevance of the comorbidity in Mexico, we considered it relevant to obtain information on the safety of TB infection therapy among patients with type 2 DM. This study aimed to assess adverse events (AEs) to TB infection treatment among patients with type 2 DM. The secondary objectives were to evaluate tolerability, adherence, proportion of treatment completion, and hepatotoxicity in patients living with type 2 DM and TB infection.

## 2. Methods

### 2.1. Study Design, Participants, and Randomization

We conducted an open-label, parallel-group, randomized, and controlled trial. We recruited consenting adults of 18 years of age or older, males and non-pregnant females, with a previous type 2 DM diagnosis and a documented positive tuberculin skin test (TST). From 24 July 2017 to 20 February 2019, we consecutively invited all patients listed in the outpatient registry according to the order assigned by the administrative department at the time of their admission to the outpatient clinic of a tertiary care center in Mexico City. If the patient accepted to participate and met the eligibility criteria, he/she was randomized. Treatment allocation was achieved using a random-number generator in ratios of 1:1 of the a priori calculated sample size (n = 403). Participants were assigned to six months of INH 300 mg/day plus pyridoxine 75 mg or three months of RIF 600 mg/day. Follow-up was extended until 24 May 2019. We excluded patients with TB disease. We screened for active disease symptoms, chest X-ray abnormalities, and mycobacteria in sputum or other appropriate samples. We ruled out pregnancy among 18–49-year-old women by a urine pregnancy test. Subjects receiving immunosuppressive therapy, with HIV infection, severe peripheral neuropathy of any cause, previous hepatic or kidney disease, drinking habit above 70 g/week (males) or 50 g/week (females), or those with known allergy to INH, RIF, or pyridoxine were not included. TST induration ≥10 mm was considered positive. Personnel at the Instituto Nacional de Salud Pública (National Institute of Public Health) generated the random allocation sequence. Study personnel enrolled and registered participants, obtained consent, verified assignment, and administered treatment at the outpatient clinic in the study hospital.

### 2.2. Procedures

The follow-up time was six months for the INH group and three months for the RIF group. Clinical and laboratory evaluations were made on days 15, 30, 60, 90 (both groups), 120, and 180 (INH group). On every visit, a Michigan Neuropathy Screening Instrument was completed [22]. Data were entered into a questionnaire, one for each visit.

At each visit, study personnel interviewed and examined the patients for adverse events, and they were instructed to contact study personnel in case new symptoms appeared between visits. Study personnel were trained to recognize, grade, evaluate, and report adverse events following a standardized protocol before the initiation of the trial. If the treating physician decided to stop treatment due to a possible treatment-related adverse event, he/she filed a report within 24 h. If treatment interruption was temporary (<48 h) or intolerance symptoms did not merit treatment interruption, they were not reported. Adverse events reports were collected until 30 days after the end of treatment. Reports comprised clinical management, laboratory results, patient response to drug withdrawal, and results of drug re-challenge if unsuccessful. The report was delivered to an adverse event manager who ensured no details were revealing which drug the patient was receiving. This person ensured that clinical information was complete. If necessary, further information was requested from the reporting physician. The event description was then transmitted to an Adverse Event Safety Panel composed of three clinical and epidemiological experts who independently evaluated the events and were blinded to the study drug. Adverse events were categorized into one of ten types: drug interaction, rash, hepatotoxicity, gastrointestinal intolerance, hematological, pregnancy, dizziness, drug-induced pancreatitis, seizure, and others. Classification of events was based on published criteria [23,24,25]. Grade 3 hepatotoxicity was defined as liver aminotransferase levels that increased to 5 to 10 or 3 to 10 times the upper limit of normality (ULN) plus compatible symptoms. Grade 4 hepatotoxicity was defined as aminotransferase levels more than ten times the ULN. We used the National Cancer Institute Common Terminology Criteria for Adverse Events for all other adverse events. Relationship to the study drug was judged as none, unlikely, possible, or probable. If individuals with adverse events were hospitalized, these same experts determined if the hospitalization was indicated for the management of the event (yes or no). In the case of panel disagreement, a simple majority was used. The panel members were asked to independently reassess if a majority was not reached. Treatment adherence was evaluated through pill count in every visit. We considered that the participant was adherent when he/she ingested ≥80% of the doses. Maximal allowed time to be off treatment was three weeks in both groups.

The presence and severity of DM complications were quantified using the Diabetes Complications Severity Index tool (DCSI) [26]. Disease severity and comorbidity were measured using the Charlson Comorbidity Index (CCI) [27,28].

### 2.3. Outcomes

We only contemplated adverse events resulting in permanent treatment cessation and considered possibly or probably related to study drugs by the Safety Panel as outcomes in the statistical analysis. The primary outcome was grade 1–2 rash, recurrent grade 2 hepatotoxicity, or grade 3–5 adverse events. We included grade 1–2 rash within our primary outcome as health personnel is usually prone to interrupt medications if a rash develops, whereas, for other mild adverse events, such as grade 1–2 hepatotoxicity, guidelines recommend continuation of treatment as these are generally transient [25]. Secondary outcomes included grade 1–4 rash, grade 3–4 hepatotoxicity, grade 3–4 hematological events, and grade 3–5 non-hepatotoxic or non-rash adverse events.

Sample size calculation. The study was designed to enroll 403 participants with a power of 81% to detect a 5% difference (6% vs. 1%) in the risk of permanent treatment interruption due to adverse drug effects at the end of treatment between groups (one-sided α level, 0.05). The expected permanent drug interruption due to the adverse impacts was based on prior Mexican data [29]. It was assumed that 5% in each group would be lost to follow-up. Due to a high rate of grade 3 or 4 hepatotoxic events leading to permanent treatment interruptions, the Adverse Event Safety Panel advised the research group to halt the study prematurely. With 68 subjects in the INH group, 63 in the RIF group, and 6 (8.8%) permanent treatment interruptions in the INH group vs. none in the RIF group, we rejected the null hypothesis with a power of 0.8 and α level of 0.05 (one-sided).

### 2.4. Statistical Analysis

Descriptive analyses were performed using frequencies and percentages for qualitative variables and median and interquartile range for continuous quantitative variables. We described adverse events resulting in permanent treatment cessation and considered possibly or probably related to study drugs according to study arm and estimated the OR and 95% Cis using unadjusted generalized linear model as an extension to the binomial family.

A multivariate Cox proportional hazards model was developed to predict the risk of grade 3 and 4 adverse events, according to the treatment arm adjusting for sociodemographic and clinical variables. Proportional hazard ratios were obtained with a confidence level of 95%. Variables that did not improve the model’s likelihood and did not affect the coefficient values were eliminated from the saturated model. Finally, the most parsimonious model was used. From this model, the proportional hazards assumption was evaluated for each of the variables and globally through the Schoenfeld residuals test. The goodness of fit of the model was evaluated using Cox-Snell residuals. Statistical calculations were performed using Stata software version 15.0.

### 2.5. Ethical Considerations

The study was approved by the Ethics Committee and Research Ethics Committee (Comités de Ética y de Ética en Investigación) with reference number 1878. Financing was received from the Mexican Council of Science and Technology (Consejo Nacional de Ciencia y Tecnología) with reference number 247582. Clinical Trials.gov registration number NCT03278483.

### 2.6. Role of the Funding Source

The study’s funder had no role in study design, data collection, data analysis, data interpretation, or report writing. The corresponding authors had full access to all the study data and had final responsibility for submitting it for publication.

## 3. Results

We studied 131 subjects; 68 were allocated to the INH group and 63 to the RIF group. Participant flow is shown in Figure 1. Of the 68 patients assigned to INH, 1 patient refused to take the drug, so 67 received the medication. Six patients dropped out of the study against the study personnel’s advice due to a grade 1 or 2 gastrointestinal intolerance grade. Nine patients met our definition of an adverse effect and were advised to interrupt medication. Of the 62 patients allocated to RIF, 3 patients refused to take the medication, so 59 received the drug. Eight patients dropped out of the study against the study personnel’s advice due to grade 1 or 2 gastrointestinal intolerance. Nine patients discontinued medication due to adverse effects, according to our definition. We analyzed 68 patients in the INH group and 62 patients in the RIF group.

Table 1 describes the characteristics of enrolled patients. The median age was 57 years (IQR 50–62 years), similar between both groups. Subjects in the INH group had a longer interval since DM diagnosis (13 years vs. ten years, *p* = 0.045). Body mass index, gender, and exposure to alcohol and drugs were equally distributed between groups. DM therapy, number of medications used, baseline liver enzymes, and glycated hemoglobin were also similar between both groups (Table 1). Kidney disease was present at similar rates in both groups, although significantly longer in the INH group (7 vs. four years, *p* = 0.049). Retinopathy was reported in 19% of subjects (25/131), also more frequent in the INH group (27.9% vs. 8.0%, *p* = 0.007). Diagnosis of cataracts was reported in 20.5% of subjects in the INH group, while it was reported in 9.5% of subjects in the RIF group, *p* = 0.079. DCSI and other comorbidities were similar between groups (Table 1). Concomitant drugs received by participants are shown in Table 2. There were no differences between groups.

The Safety Panel judged that 18 events resulting in permanent treatment cessation were possibly or probably related to study drugs, 9 (13.23%) in the INH group and 9 (14.51%) in the RIF group; frequency was not significantly different between groups, Table 3. However, when categories of adverse events were analyzed, all hepatotoxicity (one recurrent and six grade 3 or 4) occurred in the INH group. In contrast, most grade 3 or 4 gastrointestinal intolerance occurred in the RIF arm.

The multivariate Cox model revealed that the treatment arm was not associated with adverse events resulting in permanent treatment cessation and was considered possibly or probably related to study drugs. In our study population, patient characteristics such as chronic kidney disease and female sex were significantly associated with this outcome. (Table 4).

Ninety-four (72.31%) patients ingested 80% or more of prescribed pills, with no difference between arms (52 (76.41%) INH group vs. 42 (67.74%) RIF group = 0.26).

## 4. Discussion

The present study aimed to assess adverse events due to TPT among patients with type 2 DM, although we prematurely halted the study based on Adverse Event Safety Panel recommendations. There was no difference between arms in the overall frequency of adverse events resulting in permanent treatment cessation and considered possibly or probably related to study drugs. However, the INH group had significantly more permanent treatment interruptions due to grade 2 recurrent or grade 3 or 4 hepatoxicity, while the RIF arm had more treatment interruptions due to grade 3 or 4 gastrointestinal intolerance. Overall adherence to treatment was 72.31%, with no difference between groups.

Previous reports on INH preventive therapy among patients with type 2 DM date from more than 50 years ago and had limited results [30,31]. Most studies have included patients with type 2 DM as part of larger study populations, and to our knowledge, there is a single study that investigated treatment completion under programmatic conditions [32].

Severe adverse events related to INH in patients with TB infection without type 2 DM have been reported with variable frequency among studies and range between 3.3% and 8.2% [33]. Several studies of TPT have been conducted in Mexico—albeit, to our knowledge, none have only included patients with type 2 DM; rates of hepatotoxicity in these studies have been similar to those described in the literature [29,34,35]. Comparison between studies is limited due to the lack of uniformity between definitions used to grade adverse events and thresholds for drug discontinuation. These effects can be classified as hepatotoxicity, gastrointestinal intolerance, hematological, and allergic such as dermatitis. Regarding hepatotoxicity, recommendations are to stop treatment in ALT elevations of ≥3× ULN or ≥5× with or without symptoms [25].

We found that the INH group presented more hepatotoxicity events than the RIF group. DM’s effect on drug metabolism has been studied for decades [36], showing in animal models and some trials in humans that poor glycemic control is associated with a higher rate of adverse events from co-administered drugs, mainly through changes in enzymes such as p450 cytochrome and protein binding dysfunction from glycosylation [37,38,39]. We observed more grade 3 and 4 hepatoxic events among participants receiving daily INH for six months than what was found by Huang and collaborators who administered daily INH for nine months (8.8% (6/68) vs. 3.2% (2/62)) to patients living with type 2 DM under programmatic conditions [32]. Our study population had a higher proportion of women, who have been described as having a higher risk of liver enzyme derangements with INH administration [25,40,41]. Our study’s frequency of severe liver toxicity in the INH group (8.8%) and none in the RIF group was similar to that seen in other case series of patients living with HIV. Hepatotoxicity has been observed in up to 18% of subjects [40]. Menzies and collaborators reported a rate of 3.7% and 0.7% among subjects taking INH and RIF, respectively, although their population was younger and <30% had comorbidities [42]. In another study, Menzies and collaborators compared over 6000 subjects, randomized to 9 months of INH versus four months of RIF, observing grade 3–4 hepatotoxicity in 1.8% vs. 0.3% RIF groups, respectively. Most of these events led to treatment interruption. The study population in this study was also younger [43].

The higher frequency of INH hepatoxicity in our study may have several explanations. Information from the Mexican Health and Nutrition Survey 2016 [44] revealed that in 2016 prevalence of previously diagnosed type 2 DM was 9.4% (95% CI 8.3–10.8%), representing approximately 6.4 million people with this condition. People are diagnosed on average at 49 years of age [45]. Our study population was almost a decade older, probably because our institution is a referral center. As it is well known, age has been associated with a higher risk of hepatotoxicity [46]. Another condition that may have favored hepatotoxicity was the pre-existence of undiagnosed and aggravated non-alcoholic fatty liver since our study population had a high rate of overweight. Finally, the median of co-ingested drugs was above polypharmacy definitions (between 3 and 6 drugs per day); therefore, drug–drug interactions might have occurred [47]. Among the co-administered drugs were antihypertensives, sulfonylureas, antidepressants, and aspirin, plus TPT, although we did not find significant differences between groups. Statin use was reported in 75%, and alcohol use in 44%.

Our trial’s treatment adherence rate was like previously reported studies [42,43]. However, adherence in our study was lower than what was observed by Huang and collaborators; they found that once-weekly INH and rifapentine for 12 weeks (3HP) or daily INH for nine months (9H) was administered to patients with type 2 DM with a completion rate of 80% or more. A greater frequency of drug interactions due to polypharmacy might explain the lower adherence in our study population.

Gastrointestinal intolerance was the most frequent adverse event in the RIF group. Rifampicin gastrointestinal intolerance is due to hypersensitivity [48,49]. In our patient population, this was probably aggravated by diabetic gastroenteropathy and interaction with other drugs [50,51].

Our study population’s median of glycated hemoglobin was 8.65%, with similar values between groups. The latter informs our study population’s poor glycemic control, which has also been observed in other low- and medium-income countries [52,53,54]. On the one hand, the latter underlies the relevance of treating this population for TB infection to avoid their increased risk of reactivation and the poor outcomes these patients show when presenting with TB disease.

The main limitation of our study was its small study sample since we halted the study by recommendation of the Adverse Event Safety Panel upon observation of adverse events. The small sample size did not allow us to evaluate the interaction of study drugs and other patients’ characteristics, such as age, sex, additional medications, or metabolic control over hepatotoxicity. Another limitation was that patients in the INH group had longer intervals since type 2 DM diagnosis than the RIF group due to the single-block and premature halting of our trial. Therefore, the higher rate of hepatoxicity might be due to DM-associated complications in the INH group. However, the DCSI values, BUN and creatinine levels, Charlson scores, other comorbidities, frequency, and interval since diagnoses of other-DM-associated complications, age, and gender were similar between both groups. Another limitation was the open-label design of our study. We used standard definitions for adverse effects based on laboratory results to minimize possible bias. Notably, members of the Adverse Event Safety Panel blindly evaluated study outcomes. Our study’s main advantage was that all participants had been previously diagnosed with type 2 DM. Consequently, subjects in our study are older than those in previous literature, making our results more widely applicable to older populations suffering from DM. Finally, our results may be generalized to similar regions with a high prevalence of type 2 DM and TB and large populations of elderly patients living with uncontrolled type 2 DM.

## 5. Conclusions

Our study shows that, in a setting where the association of type 2 DM and TB constitutes a severe public health problem, TPT with INH is not safe enough to be considered a universal indication to patients with type 2 DM and TBI infection. On the other hand, the frequency of gastrointestinal intolerance among the group receiving RIF also precludes adherence among patients receiving this drug. Previous considerations of the potential risks of hepatotoxicity due to polypharmacy and comorbidities, as well as gastric intolerance due to polypharmacy and preexistent gastric disease related to type 2 DM, are both confirmed in our study. These results underline the need to search for alternatives to TPT in type 2 DM patients with better safety profiles, such as rifapentine.

## Figures and Tables

**Figure 1 microorganisms-11-01917-f001:**
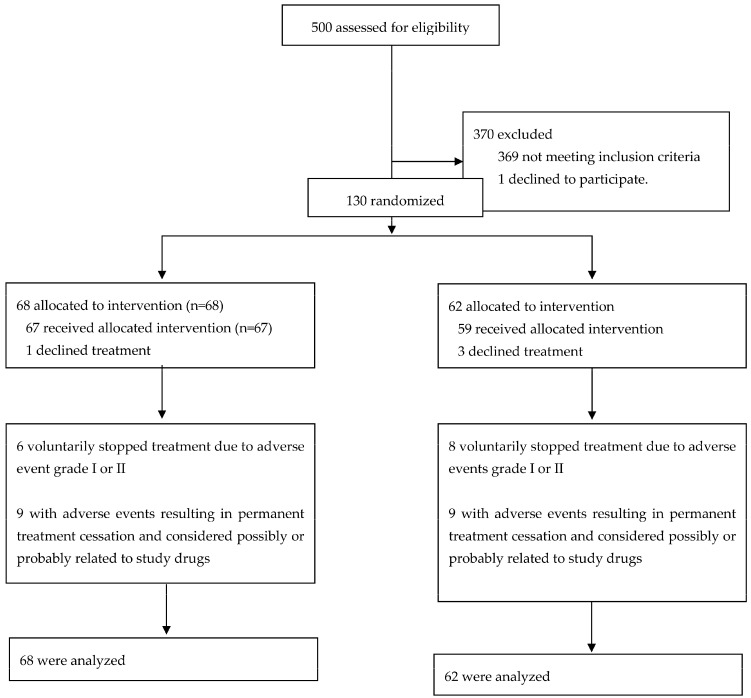
Flow Diagram.

**Table 1 microorganisms-11-01917-t001:** Baseline characteristics in both treatment groups.

Variable	Total N = 130 (%)	Isoniazid n = 68 (%)	Rifampicin n = 62 (%)	*p* Value *
Age (Median, IQR)	57 (50–62)	57 (52–62)	57 (48–62)	0.464
Men	54 (41.2)	24 (35.2)	30 (47.0)	0.152 **
BMI (kg/m^2^)	28.1 (25.6–31.6)	27.7 (25.6–31.2)	28.7 (25.6–31.7)	0.622
Time since DM diagnosis (years)	11 (4–19)	13 (5–23)	10 (3.5–15.5)	0.045
Alcohol drinking	58 (44.2)	29 (42.6)	29 (45.2)	0.697 **
Cigarette smoking	60 (46.5)	29 (42.6)	31 (50.0)	0.350 **
Oral glucose-lowering drugs	128 (98.4)	68 (100)	60 (96.8)	0.139 **
Insulin	73 (55.7)	42 (61.8)	31 (49.2)	0.148 **
Years of insulin use	7 (4–12)	8 (4–14)	6.5 (4–11)	0.673
Number of drugs taken	6 (4–7)	6 (5–8)	5 (4–7)	0.171
Chronic kidney disease	28 (22.1)	15 (22)	13 (21.0)	0.982
Time since diagnosis (years)	7 (6–7)	7 (7–8.5)	4 (2–6)	0.049
Retinopathy	24 (19)	19 (27.9)	5 (8.0)	0.007
Time since diagnosis of retinopathy (years)	3 (2–9)	3.5 (2–9)	2.5 (1–6)	0.349
Foot ulcer or amputation	5 (4.5)	4 (5.8)	1 (1.6)	0.459
Ischemic heart disease	4 (3.8)	3 (4.4)	1 (1.6)	0.712
Neuropathy	25 (19.8)	14 (20.5)	11 (17.7)	0.825
Diabetes Complications Severity Index	1 (1–2)	1 (1–3)	1 (0–2)	0.064
Comorbidities				
Hypertension	66 (51.9)	39 (57.3)	27 (45.9)	0.194
Time since diagnosis of hypertension	10.5 (4.5–18)	11 (6–18)	10 (4–18)	0.557
Obesity	39 (29.7)	18 (26.4)	21 (33.3)	0.391
Time since diagnosis of obesity	15 (10–28)	18 (12–28)	13 (7–28.5)	0.464
Dyslipidemia	85 (64.8)	42 (61.7)	43 (68.2)	0.437
Depressive symptoms	34 (25.9)	17 (25)	17 (26.9)	0.796
Gastrointestinal symptoms	6 (4.5)	3 (4.4)	3 (4.7)	0.924
Osteoarthritis	13 (9.9)	8 (11.7)	5 (7.9)	0.464
Cataratacts	20 (15.2)	14 (20.5)	6 (9.5)	0.079
Glaucoma	6 (4.5)	5 (7.3)	1 (1.5)	0.115
Charlson Index score	3 (2–4)	3 (2–4)	3 (2–4)	0.545
Baseline Laboratory parameters				
Alanine aminotransferase (U/L) median (IQR)	18.9 (14.2–23.6)	19 (14–26.9)	18.6 (14.7–21.9)	0.778
Aspartate aminotransferase (U/L) median (IQR)	18 (15–22)	19 (15.5–23.5)	18 (15–21)	0.538
Total bilirubin (mg/dL) median (IQR)	0.53 (0.43–0.63)	0.53 (0.43–0.62)	0.53 (0.43–0.64)	0.661
Glycated hemoglobin (median, IQR)	8.2 (6.6–10)	8.2 (6.8–9.7)	8 (6.4–10.2)	0.933

* Mann–Whitney test; ** Χ^2^ test.

**Table 2 microorganisms-11-01917-t002:** Concomitant drugs received by participants, according to study arm.

Concomitant Drug	Total N = 130 (%)	Isoniazid n = 68 (%)	Rifampicin n = 62 (%)	*p* Value **
Metformin	123 (93.9)	64 (94.1)	59 (93.6)	0.911
Sulfonylureas	17 (13)	7 (10.3)	10 (15.9)	0.342
Sitagliptin	13 (9.9)	8 (11.7)	5 (7.9)	0.464
SLGT2 inhibitors *	15 (11.5)	9 (13.2)	6 (9.5)	0.505
Statins	98 (74.8)	53 (77.9)	45 (71.4)	0.391
Fibrates	48 (36.6)	21 (30.9)	27 (42.9)	0.155
Antidepressants	20 (15.3)	10 (14.7)	10 (15.9)	0.853
Antihypertensive agents	81 (61.8)	44 (64.7)	37 (58.7)	0.482
Aspirin	70 (53.4)	36 (52.9)	34 (54)	0.906
Other	76 (58)	40 (58.8)	36 (57.1)	0.846

* Dapagliflozin or empagliflozin; ** Χ^2^ test.

**Table 3 microorganisms-11-01917-t003:** Adverse events resulting in permanent treatment cessation and considered possibly or probably related to study drugs.

Outcome	Total N = 130	Isoniazid n = 68	Rifampicin n = 62	Relative Risk (95% CI) *
Primary outcome				
Recurrent grade 2 hepatotoxicity or grade 3–4 adverse events	18 (13.84)	9 (13.23)	9 (14.51)	1.09 (0.47 to 2.59)
Secondary Outcomes				
Recurrent grade 2 hepatotoxicity	1 (0.76)	1 (1.47)	0	-----
Grade 3–4 hepatotoxicity	6 (4.61)	6 (8.82)	0	-----
Grade 3–4 gastrointestinal intolerance	11 (8.46)	2 (2.94)	9 (14.51)	4.93 (1.11 to 21.96)

* Unadjusted generalized linear model.

**Table 4 microorganisms-11-01917-t004:** Multivariate Cox model for the association of clinical and sociodemographic factors with adverse events resulting in permanent treatment cessation and considered possibly or probably related to study drugs.

Characteristics	Adverse Events III-IV Related to Therapy Hazard Ratio	95% Confidence Interval	*p* Value
Chronic kidney disease	6.44	1.56–26.58	0.010
Female	4.53	1.25–16.39	0.021
Alcohol usage	2.02	0.70–5.84	0.196
Age	1.03	0.97–1.09	0.359
Isoniazid arm	0.62	0.24–1.57	0.311

## Data Availability

The data presented in this study are available in the Appendix A.

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
