# Peer review of "Safety and Tolerability of Six Months of Isoniazid Plus Pyridoxine or Three Months of Rifampicin for Tuberculosis among Subjects with Diabetes Mellitus: A Randomized Trial"

_microorganisms, 2023, doi:10.3390/microorganisms11081917_

Round 1

Reviewer 1 Report

Dear Authors,

The aim of your manuscript was to investigate the adverse effects which may occur during two drug regimens for tuberculosis treatment, in groups of patients with diabetes. 

After carefully reading the manuscript, I have some comments and suggestions:

1. in the abstract you say that a drug regimen administered was isoniazid with pyridoxine, but pyridoxine is not mentioned in the title. Why not?

2. erase the dot after the title

3. explain what LTBI abbreviation stands for

4. you did not precise the type of DM. this is important, in order to know the organs affected by this disease and also to know the antidiabetic medication administered

5. did you exclude the adverse effects of the antidiabetic drugs administered? for example, for diabetes type 2, there are oral drugs that have severe gastrointestinal effects. These have to be excluded, because you mention that the gastrointestinal disorders where the main adverse effects in the group receiving rifampicin.

6. in the Introduction part, you mention the relationship between DM and TB. You have to detail. Plus, you say that "DM triples the risk of active tuberculosis". Why? What are the mechanisms? It is not clear. Please detail, it is mandatory for this type of affirmation.

7. reduce Figure 1 and also use the same letter font as in text

8. Separate the Discussions from the Conclusions and detail the conclusions.

There are minor English mistakes, such as: elimination of "the" before WHO in Introduction

Author Response

Answers to Reviewer no. 1

  1. In the abstract you say that a drug regimen administered was isoniazid with pyiridoxine, but pyridoxine is not mentioned in the title. Why not?

Answer: We have modified the title accordingly.

  1. Erase the dot after the title

Answer: We have modified the text accordingly.

  1. Explain what LTBI abbreviation stands for

Answer: We have explained accordingly.

  1. You did not precise the type of DM. this is important, in order to know the organs affected by this disease and also to know the antidiabetic medication administered

Answer: We have modified the text accordingly.

  1. Did you exclude the adverse effects of the antidiabetic drugs administered? for example, for diabetes type 2, there are oral drugs that have severe gastrointestinal effects. These have to be excluded, because you mention that the gastrointestinal disorders where the main adverse effects in the group receiving rifampicin.

Answer: Our definition of the primary outcome (adverse events resulting in permanent treatment cessation and considered possibly or probably related to study drugs by the Safety Panel) excludes adverse effects due to only antidiabetic drugs. However, antidiabetic drugs may have potentiated adverse effects of INH or RIF. We have added a table (table 2) explaining all the medications for treating type 2 diabetes that participants were receiving and have compared them according to the study arm. There were no differences between arms. Therefore, the adverse effects of antidiabetic drugs were equally distributed between study groups. The revision of the Safety Panel that defined adverse effects as only those possibly or probably related to study drugs and the fact that antidiabetic drugs were equally distributed between groups decreased to a minimal the bias that antidiabetic medications might have introduced.

  1. In the Introduction part, you mention the relationship between DM and TB. You have to detail. Plus, you say that "DM triples the risk of active tuberculosis." Why? What are the mechanisms? It is not clear. Please detail, it is mandatory for this type of affirmation.

Answer: We have added a paragraph detailing the studies that have found the causal association between diabetes and tuberculosis. We have also added studies explaining the immunology of the interaction of diabetes and tuberculosis.

  1. Reduce Figure 1 and also use the same letter font as in text

Answer: We have modified the figure accordingly.

  1. Separate the Discussions from the Conclusions and detail the conclusions

Answer: We have modified the text accordingly.

9. There are minor English mistakes, such as: elimination of "the" before WHO in Introduction

Answer: We have edited the English language.

Reviewer 2 Report

The manuscript describes in detail use of isoniazid and Rifapentin in patients with diabetes (DM) and LTBI. The adverse effects were significantly high and thus the study needed to be terminated prematurely without continuing for the entire duration of the treatment.  This has been shown clearly with the data collected and presented and indicates that alternative drugs are needed for the treatment of DM patients. This is an important finding although an incomplete study.  Were these patients also on any antiDM treatment? (Metformin)

writing can be improved, there are many incomplete sentences, isoniazid is written as NIH please correct it as INH

"The report was delivered to an adverse event manager who ensured there were no details revealing which drug the pa-tient was receiving and ensured that the information was complete. "

Not sure what this means

Author Response

Answers to the Reviewer no. 2

  1. Writing can be improved, there are many incomplete sentences, isoniazid is written as NIH please correct it as INH.

Answer: We have modified the text accordingly.

  1. "The report was delivered to an adverse event manager who ensured there were no details revealing which drug the patient was receiving and ensured that the information was complete."

Not sure what this means

Answer: We have modified the text accordingly.